# FULLY DIFFERENTIABLE MODEL DISCOVERY

## ABSTRACT

Model discovery aims at autonomously discovering differential equations under-lying a dataset. Approaches based on Physics Informed Neural Networks (PINNs) have shown great promise, but a fully-differentiable model which explicitly learns the equation has remained elusive. In this paper we propose such an approach by integrating neural network-based surrogates with Sparse Bayesian Learning (SBL). This combination yields a robust model discovery algorithm, which we showcase on various datasets. We then identify a connection with multitask learn-ing, and build on it to construct a Physics Informed Normalizing Flow (PINF). We present a proof-of-concept using a PINF to directly learn a density model from single particle data. Our work expands PINNs to various types of neural network architectures, and connects neural network-based surrogates to the rich field of Bayesian parameter inference.

## 1 INTRODUCTION

Many physical, chemical and biological systems can be modelled with (partial) differential equa-tions. They capture a system's interactions, scales and conserved quantities in an interpretive man-ner. Unfortunately, manually deriving these equations from first principles is a time-consuming process and requires expert knowledge of the underlying dynamics. The last few years have seen a rising interest in automating this process, also known as *model discovery*. As the model space is exponentially large, a popular approach is to select a large set of candidate terms (features) and perform sparse regression on these features, effectively turning model discovery into variable selec-tion (Brunton et al., 2016; Rudy et al., 2017). A uniquely challenging aspect of discovering PDEs is that many of the candidate features contain higher-order derivatives, which are challenging to calcu-late accurately using numerical differentiation. This essentially limited model discovery to densely sampled datasets with low noise levels.

Various works have explored the use of neural networks to generate a surrogate of the data (Rasheed et al., 2020), and perform model discovery on this surrogate instead (Berg & Nyström, 2019; Both et al., 2019). By using a neural network $g$ to learn the data $u$ such that $u \approx g(x,t)$, the network denoises the data, while automatic differentiation can be used to accurately calculate the (higher-) order derivatives of $u$ used in the candidate features. These approaches show significant improve-ments when the neural network is constrained to solutions allowed by the candidate features. This essentially yields a Physics Informed Neural Network (PINN) (Raissi et al., 2017), with the impor-tant distinction that the form of the constraint, i.e. which terms of the candidate features are active and make up the underlying equation, is also learned. As it is composed of all candidate features, the constraint itself is prone to overfitting, and the discovered equation will contain more terms than re-quired. Applying $\ell_1$ regularisation can alleviate this problem, but raises the question of how strongly to apply it. Alternatively, inactive terms can be pruned from the constraint by applying a mask, but this is a non-differentiable operation, and training the network does not take sparsity into account. The open question then is how to optimally apply a constraint, which itself is learned and sensitive to overfitting, all the while remaining fully differentiable.

In this work we introduce a fully differentiable model discovery algorithm consisting of a neural-network based surrogate with a constraint based on Sparse Bayesian Learning (SBL). We summarise our contributions as follows:

- We show how Bayesian parameter inference methods can be used as a constraint in a PINN. Specifically, we use Sparse Bayesian Learning (SBL) to create a fully-differentiable, robust model discovery algorithm and showcase this on various datasets.

- We identify a connection with multitask learning using uncertainty and exploit this to generalise PINNs to probabilistic surrogates. We introduce a conditional normalizing flow constrained by SBL, a so called Physics Informed Normalizing Flow (PINF).

- We present a proof-of-concept where PINF learns a time-dependent density from unlabelled single-particle data, allowing the constraint to discover a density model directly from single particle data.

## 2 BACKGROUND

**Model discovery with sparse regression**   Model discovery aims to discover the PDE from a large set of $M$ candidate features $\{u, u_{xx}, uu_x, \ldots\}$. Assuming the underlying equation can be written as a linear combination of the candidate features, model discovery can be approached as a regression problem (Brunton et al., 2016) by solving

$$\hat{\boldsymbol{\xi}} = \min_{\boldsymbol{\xi}} \|\partial_t \boldsymbol{u} - \boldsymbol{\Theta}\boldsymbol{\xi}\|^2 + \lambda R(\boldsymbol{\xi}), \tag{1}$$

where $\boldsymbol{\Theta} \in \mathcal{R}^{N \times M}$ contains all candidate features, $\boldsymbol{\xi} \in \mathcal{R}^M$ the unknown coefficient vector and $R(\boldsymbol{\xi})$ some sparsity-promoting penalty; the number of candidate features is typically much larger than the number of terms in the underlying equation. The main challenge of discovering the underlying equation using this approach is dealing with large, possible correlated errors in the features containing derivatives; using numerical differentiation to calculate these higher-order derivatives accurately from noisy and sparse data is extremely challenging, even after denoising. One line of work has focused on constructing more robust and sparser approaches to solving eq. 1, for example SR3 (Zheng et al., 2019) or using stability criteria (Maddu et al., 2020). Alternatively, several works (Both et al., 2019; Berg & Nyström, 2019) have explored the use of neural networks to create a surrogate of the data and perform model discovery on this surrogate instead. Automatic differentiation can then be used to calculate the derivatives, yielding much more accurate features.

**PINNs**   Physics Informed Neural Networks (PINNs) (Raissi et al., 2017) have become a very popular method to either solve a differential equation or perform parameter inference with neural networks. Here we focus on parameter inference and consider a (noisy) dataset $\{\boldsymbol{u}_i, \boldsymbol{x}_i, t_i\}_{i=1}^N$, governed by a differential equation of form $\partial_t \boldsymbol{u} = \boldsymbol{X}\boldsymbol{w}$ with $\boldsymbol{X}$ the terms of the equation and $\boldsymbol{w}$ the unknown parameters. PINNs infer $\boldsymbol{w}$ by using a neural network to approximate $\boldsymbol{u}$, and constrain the network to the given differential equation by minimising

$$\mathcal{L}_{\mathrm{PINN}}(\boldsymbol{\theta}, \boldsymbol{w}) = \frac{1}{N}\sum_{i=1}^N \|\hat{\boldsymbol{u}}_i - \boldsymbol{u}_i\|^2 + \frac{\lambda}{N}\sum_{i=1}^N \|\partial_t \hat{\boldsymbol{u}}_i - \boldsymbol{X}_i\boldsymbol{w}\|^2. \tag{2}$$

Here $\hat{\boldsymbol{u}}_i = g_{\boldsymbol{\theta}}(\boldsymbol{x}_i, t_i)$ is the prediction of the neural network and $\lambda$ sets the strength of the regularisation. The constraint ensures the network approximates the data $\boldsymbol{u}$ consistently with the given differential equation, and terms containing derivatives can be calculated using automatic differentiation. These two features make PINNs especially useful in noisy and sparse datasets.

**Model discovery with PINNs**   PINNs can easily be adapted to perform model discovery by replacing the given differential equation $\boldsymbol{X}$ with a larger set of candidate features $\boldsymbol{\Theta}$. Additionally, a mask $\boldsymbol{m}$ is applied to the coefficients, yielding as loss function,

$$\mathcal{L}_{\mathrm{MD}}(\boldsymbol{\theta}, \boldsymbol{\xi}) = \frac{1}{N}\sum_{i=1}^N \|\hat{\boldsymbol{u}}_i - \boldsymbol{u}_i\|^2 + \frac{\lambda}{N}\sum_{i=1}^N \|\partial_t \hat{\boldsymbol{u}}_i - \boldsymbol{\Theta}_i(\boldsymbol{m} \odot \boldsymbol{\xi})\|^2$$
$$= \mathcal{L}_{\mathrm{fit}}(\boldsymbol{\theta}) + \lambda \mathcal{L}_{\mathrm{reg}}(\boldsymbol{\theta}, \boldsymbol{\xi}). \tag{3}$$

The mask $\boldsymbol{m}$ describes which terms feature in the equation, and hence the form of the constraint; this approach can be interpreted as a PINN in which the constraint is also learned. The mask is updated periodically by some sparse regression technique, and as terms are pruned, the constraint becomes

stricter, preventing overfitting of the constraint itself and improving the approximation of the network, boosting performance significantly (Both et al., 2021). However, the non-differentiability of the mask can lead to issues during training, for example when it is updated at the wrong time, or when the wrong terms are accidentally pruned. Our goal here is thus to construct an approach where the mask $M$ is learned together with the networks parameters, while still maintaining the benefits of iteratively refining the approximation: a fully-differentiable model discovery algorithm.

**Removing free variables**   Training with eq. 3 optimises two sets of parameters: the network parameters $\theta$ and the coefficients $\xi$. Typically both are minimised together using gradient descent, but the optimisation of $\xi$ can be performed analytically (Both et al., 2021). Given a configuration of the network parameters $\theta^*$, the minimisation over $\xi$ is a regression problem as given by eq. 1 and can be solved exactly. Referring to this solution as the maximum likelihood estimate $\xi_{\text{MLE}}$, we define a loss function $\tilde{\mathcal{L}}_{\text{MD}}(\theta) \equiv \mathcal{L}_{\text{MD}}(\theta, \xi_{\text{MLE}})$, which optimises only the network parameters $\theta$ using gradient descent. This significantly speeds up convergence and reduces the variance of the discovered coefficients across initialisations. We shall adopt this approach in the rest of the paper, and define the convention of $\tilde{\mathcal{L}}$ denoting the loss function $\mathcal{L}$ with the independent variables calculated analytically.

## 3   FULLY DIFFERENTIABLE MODEL DISCOVERY

Our goal is to create a fully-differentiable model discovery algorithm, which, considering eq. 3, requires making the mask $m$ differentiable. Differentiable masking is challenging due to the binary nature of the problem, and instead *we relax the application of the mask to a regularisation problem*. Specifically, we propose to use Sparse Bayesian Learning (Tipping, 2001) to select the active features and act as constraint. We start this section by reviewing SBL and how it can be used for differentiable variable selection, next show to integrate it in PINNs and finally introduce Physics Informed Normalizing Flows.

### 3.1   DIFFERENTIABLE MASKING WITH SBL

**Sparse Bayesian Learning**   Sparse Bayesian Learning (SBL) (Tipping, 2001) is a Bayesian approach to regression yielding sparse results. SBL defines a hierarchical model, starting with a Gaussian likelihood with noise precision $\beta \equiv \sigma^{-2}$, and a zero-mean Gaussian with precision $\alpha_j$ on each component $\xi_j$ as prior,

$$p(\partial_t \hat{u};\ \Theta, \xi, \beta) = \prod_{i=1}^{N} \mathcal{N}(\partial_t \hat{u}_i;\ \Theta_i \xi, \beta^{-1}), \tag{4}$$

$$p(\xi;\ A) = \prod_{j=1}^{M} \mathcal{N}(\xi_j;\ 0, \alpha_j^{-1}), \tag{5}$$

with $\partial_t \hat{u} \in \mathcal{R}^N$, $\Theta \in \mathcal{R}^{N \times M}$, $\xi \in \mathcal{R}^M$, and we have defined $A \equiv \text{diag}(\alpha)$. The posterior distribution of $\xi$ is a Gaussian with mean $\mu$ and covariance $\Sigma$,

$$\begin{aligned} \Sigma &= (\beta \Theta^T \Theta + A)^{-1} \\ \mu &= \beta \Sigma \Theta^T \partial_t \hat{u}. \end{aligned} \tag{6}$$

Many of the terms in $A$ will go to infinity when optimised, and correspondingly the prior for term $j$ becomes a delta peak. We are thus certain that that specific term is inactive and can be pruned from the model. This makes SBL a very suitable choice for model discovery, as it gives a rigorous criterion for deciding whether a term is active or not. Additionally it defines hyper-priors over $\alpha$ and $\beta$,

$$\begin{aligned} p(\alpha) &= \prod_{j=1}^{M} \Gamma(\alpha_j;\ a, b) \\ p(\beta) &= \Gamma(\beta;\ c, d) \end{aligned} \tag{7}$$

The inference of $\boldsymbol{A}$ and $\beta$ cannot be performed exactly, and SBL uses type-II maximum likelihood to find the most likely values of $\hat{\boldsymbol{A}}$ and $\hat{\beta}$ by minimising the negative log marginal likelihood [1],

$$\mathcal{L}_{\text{SBL}}(\boldsymbol{A}, \beta) = \frac{1}{2} \left[ \beta \left\| \boldsymbol{u}_t - \boldsymbol{\Theta}\boldsymbol{\mu} \right\|^2 + \boldsymbol{\mu}^T \boldsymbol{A} \boldsymbol{\mu} - \log|\boldsymbol{\Sigma}| - \log|\boldsymbol{A}| - N \log \beta \right] -$$
$$\sum_{j=1}^{M} (a \log \alpha_j - b\alpha_j) - c \log \beta + d\beta, \quad (9)$$

using an iterative method (see Tipping (2001)).

**Continuous relaxation** The marginal likelihood also offers insight how the SBL provides differentiable masking. Considering only the first two terms of eq. 9,

$$\beta \left\| \boldsymbol{u}_t - \boldsymbol{\Theta}\boldsymbol{\mu} \right\|^2 + \boldsymbol{\mu}^T \boldsymbol{A} \boldsymbol{\mu} \quad (10)$$

we note that the SBL essentially applies a coefficient-specific $\ell_2$ penalty to the posterior mean $\boldsymbol{\mu}$. If $A_j \to \infty$, the corresponding coefficient $\mu_j \to 0$, pruning the variable from the model. Effectively, the SBL replaces the discrete mask $m_j \in \{0, 1\}$ by a continuous regularisation $A_j \in (0, \infty]$, and we thus refer to our approach as *continuous relaxation*.

## 3.2 SBL-CONSTRAINED PINNS

**Model** To integrate SBL as a constraint in PINNs (similar to eq. 2), we place a Gaussian likelihood on the output of the neural network,

$$\hat{u} : p(\boldsymbol{u};\ \hat{\boldsymbol{u}}, \tau) = \prod_{i=1}^{N} \mathcal{N}(\boldsymbol{u}_i;\ \hat{\boldsymbol{u}}_i,\ \tau^{-1}), \quad (11)$$

and define a Gamma hyper prior on $\tau$, $p(\tau) = \Gamma(\tau;\ e, f)$, yielding the loss function,

$$\mathcal{L}_{\text{data}}(\boldsymbol{\theta}, \tau) = \frac{1}{2} \left[ \tau \left\| \boldsymbol{u} - \hat{\boldsymbol{u}} \right\|^2 - N \log \tau \right] - e \log \tau + f\tau. \quad (12)$$

Assuming the likelihoods factorise, i.e. $p(\boldsymbol{u}, \boldsymbol{u}_t;\ \hat{\boldsymbol{u}},\ \boldsymbol{\Theta},\ \boldsymbol{\xi}) = p(\boldsymbol{u};\ \hat{\boldsymbol{u}}) \cdot p(\boldsymbol{u}_t; \boldsymbol{\Theta},\ \boldsymbol{\xi})$, SBL can be integrated as a constraint in a PINN by simply adding the two losses given by eq. 9 and eq. 12,

$$\mathcal{L}_{\text{SBL-PINN}}(\boldsymbol{\theta}, \boldsymbol{A}, \tau, \beta) = \mathcal{L}_{\text{data}}(\boldsymbol{\theta}, \tau) + \mathcal{L}_{\text{SBL}}(\boldsymbol{\theta}, \boldsymbol{A}, \beta) \quad (13)$$

Our approach does not rely on any specific property of the SBL, and thus generalises to other Bayesian regression approaches.

**Training** The loss function for the SBL-constrained PINN contains three variables which can be exactly minimised, and denote these as $\hat{\boldsymbol{A}}, \hat{\tau}$ and $\hat{\beta}$. With these values, we introduce $\tilde{\mathcal{L}}_{\text{SBL-PINN}}(\boldsymbol{\theta}) \equiv \mathcal{L}_{\text{SBL-PINN}}(\boldsymbol{\theta}, \hat{\boldsymbol{A}}, \hat{\tau}, \hat{\beta})$ and note that we can further simplify this expression as the gradient of the loss with respect to these variables is zero. For example, $\nabla_{\boldsymbol{\theta}}\mathcal{L}(\hat{\boldsymbol{A}}) = \nabla_{\boldsymbol{A}}\mathcal{L} \cdot \nabla_{\boldsymbol{\theta}}\boldsymbol{A}|_{\boldsymbol{A}=\hat{\boldsymbol{A}}} = 0$, as $\nabla_{\boldsymbol{A}}\mathcal{L}|_{\boldsymbol{A}=\hat{\boldsymbol{A}}} = 0$. Thus, keeping only terms directly depending on the neural network parameters $\boldsymbol{\theta}$ yields,

$$\tilde{\mathcal{L}}_{\text{SBL-PINN}}(\boldsymbol{\theta}) = \frac{\hat{\tau}}{2} \left\| \boldsymbol{u} - \hat{\boldsymbol{u}} \right\|^2 + \frac{\hat{\beta}}{2} \left\| \boldsymbol{u}_t - \boldsymbol{\Theta}\boldsymbol{\mu} \right\|^2 + \boldsymbol{\mu}^T \hat{\boldsymbol{A}} \boldsymbol{\mu} - \log|\boldsymbol{\Sigma}|$$
$$= \frac{N\hat{\tau}}{2} \underbrace{\left[ \mathcal{L}_{\text{fit}}(\boldsymbol{\theta}) + \frac{\hat{\beta}}{\hat{\tau}} \mathcal{L}_{\text{reg}}(\boldsymbol{\theta}, \boldsymbol{\mu}) \right]}_{=\mathcal{L}_{\text{PINN}}(\boldsymbol{\theta}, \mu)} + \boldsymbol{\mu}^T \hat{\boldsymbol{A}} \boldsymbol{\mu} - \log|\boldsymbol{\Sigma}| \quad (14)$$

---

[1] Neglecting the hyper-prior, this loss function can also written more compactly as

$$\mathcal{L}_{\text{SBL}}(\boldsymbol{A}, \beta) = \log |\boldsymbol{C}| + \partial_t \boldsymbol{u}^T \boldsymbol{C}^{-1} \partial_t \boldsymbol{u}, \quad \boldsymbol{C} = \beta^{-1}\boldsymbol{I} + \boldsymbol{\Theta}\boldsymbol{A}^{-1}\boldsymbol{\Theta}^T, \quad (8)$$

but the format we use provides more insight how SBL provides differentiable masking.

where in the second line we have rewritten the loss function in terms of a classical PINN with relative regularisation strength $\lambda = \hat{\beta}/\hat{\tau}$ and coefficients $\boldsymbol{\xi} = \boldsymbol{\mu}$. Contrary to a PINN however, the regularisation strength is inferred from the data, and the coefficients $\boldsymbol{\mu}$ are inherently sparse.

An additional consequence of $\nabla_{\boldsymbol{\theta}} \mathcal{L}(\hat{\boldsymbol{A}}, \hat{\beta}, \hat{\tau}) = 0$ is that our method does not require backpropagating through the solver. While such an operation could be efficiently performed using implicit differentiation (Bai et al., 2019), our method requires solving an iterative problem only in the forward pass. During the backwards pass the values obtained during the forward pass can be considered constant.

**Connection with multitask learning**  Considering eq. 14, we note the resemblance to multitask learning using uncertainty, introduced by Cipolla et al. (2018). Given a set of objectives, the authors propose placing a Gaussian likelihood on each objective so that each task gets weighed by its uncertainty. The similarity implies that we are essentially reinterpreting PINNs as Bayesian or hierarchical multi-task models.

## 3.3 Physics Informed Normalizing Flows

Having redefined the PINN loss function (eq. 2) in terms of likelihoods (i.e. eq. 14) allows to introduce a PINN-type constraint to any architecture with a probabilistic loss function. In this section we introduce an approach with normalizing flows, called Physics Informed Normalizing Flows (PINFs). As most physical equations involve time, we first shortly discuss how to construct a time-dependent normalizing flow. We show in the experiments section how PINFs can be used to directly infer a density model from single particle observations.

**Conditional Normalizing Flows**  Normalizing flows construct arbitrary probability distributions by applying a series of $K$ invertible transformations $f$ to a known probability distribution $\pi(\boldsymbol{z})$,

$$\boldsymbol{z} = f_K \circ \ldots \circ f_0(\boldsymbol{x}) \equiv g_{\boldsymbol{\theta}}(\boldsymbol{x})$$

$$\log p(\boldsymbol{x}) = \log \pi(\boldsymbol{z}) + \sum_{k=1}^{K} \log \left| \det \frac{\partial f_k(\boldsymbol{z})}{\partial d\boldsymbol{z}} \right|, \tag{15}$$

and are trained by minimising the negative log likelihood, $\mathcal{L}_{\mathrm{NF}} = -\sum_{i=1}^{N} \log p(\boldsymbol{x})$. Most physical processes yield time-dependent densities, meaning that the spatial axis is a proper probability distribution with $\int p(\boldsymbol{x}, t) d\boldsymbol{x} = 1$. Contrarily, this is not valid along the temporal axis, as $\int p(\boldsymbol{x}, t) dt = f(\boldsymbol{x})$. To construct PINFs, we first require a Conditional Normalizing Flow capable of modelling such time-dependent densities. Instead of following the method of Both & Kusters (2019), which modifies the Jacobian, we employ time-dependent hyper-network. This hyper-network $h$ outputs the flow parameters $\boldsymbol{\theta}$, and is only dependent on time, i.e. $\boldsymbol{\theta} = h(t)$, thus defining a time-dependent normalizing flow as $\boldsymbol{z} = g_{h(t)}(\boldsymbol{x})$.

**PINFs**  Conditional normalizing flows yield a continuous spatio-temporal density, and the loss function of a PINF is defined as simply adding the SBL-loss to that of the normalizing flow, yielding

$$\tilde{\mathcal{L}}_{\mathrm{PINF}}(\boldsymbol{\theta}) = \mathcal{L}_{\mathrm{NF}}(\boldsymbol{\theta}) + \frac{N\hat{\beta}}{2} \mathcal{L}_{\mathrm{reg}}(\boldsymbol{\theta}, \boldsymbol{\mu}) + \boldsymbol{\mu}^T \hat{\boldsymbol{A}} \boldsymbol{\mu} - \log|\boldsymbol{\Sigma}|. \tag{16}$$

## 4 Experiments

We now show several experiments illustrating our approach. We start this section by discussing choice of hyperprior, followed by a benchmark on several datasets and finally a proof-of-concept with physics-informed normalizing flows.

## 4.1 Choosing prior

The loss function for the SBL constrained approach contains several hyper-parameters, all defining the (hyper-) priors on respectively $\boldsymbol{A}$, $\beta$ and $\tau$. We set uninformed priors on $\boldsymbol{A}$ and $\beta$, $a = b =$

$e = f = 1e^{-6}$, but those on $\beta$, the precision of the constraint, must be chosen more carefully. Figure 1 illustrates the learning dynamics on a dataset of the Korteweg-de Vries equation [2] when the $\beta$ hyperprior is uninformed, i.e. $c = d = 1e^{-6}$. Observe that the model fails to learn the data, while almost immediately optimising the constraint. We explain this behaviour as a consequence of our assumption that the likelihoods factorise, which implies the two tasks of learning the data and applying the constraint are independent. Since the constraint contains much more terms than required, it can fit a model with high precision to any output the neural network produces. The two tasks then are not independent but conditional: a high precision on the constraint is warranted only if the data is reasonably approximated by the neural network. To escape the local minimum observed in figure 1, we couple the two tasks by making the hyper-prior on $\beta$ dependent on the performance of the fitting task.

**Dynamic prior**  Our starting point is the update equation for $\beta$ (see Tipping (2001) for details),

$$\hat{\beta} = \frac{N - M + \sum_i \boldsymbol{\alpha}_i \boldsymbol{\Sigma}_{ii} + 2c}{N\mathcal{L}_{\text{reg}} + 2d} \qquad (17)$$

We typically observe good convergence of normal PINNs with $\lambda = 1$, and following this implies $\hat{\beta} \approx \hat{\tau}$, and similarly $\mathcal{L}_{\text{reg}} \to 0$ as the model converges. Assuming $N \gg M + \sum_i \alpha_i \Sigma_{ii}$, we have

$$\hat{\tau} \approx \frac{N + 2c}{2d}, \qquad (18)$$

which can be satisfied with $c = N/2$, $d = n/\hat{\tau}$. Figure 1 shows that with this dynamic prior the SBL constrained PINN does not get trapped in a local minimum and learns the underlying data. We hope to exploit multitask learning techniques to optimize this choice in future work.

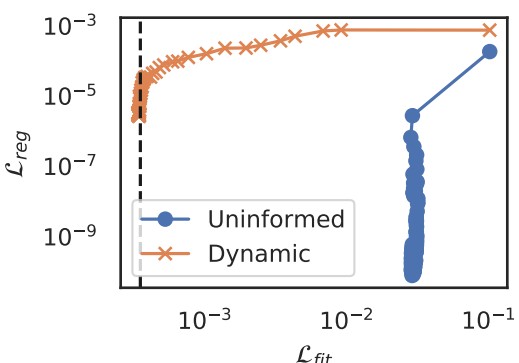

Figure 1: Regression loss as a function of fitting loss during training, comparing an uninformed prior with a dynamic prior.

## 4.2 EXPERIMENTS

We present three experiments to benchmark our approach. We first study the learning dynamics in-depth on a solution of the Korteweg- de Vries equation, followed by a robustness study of the Burgers equation, and finally show the ability to discover the chaotic Kuramoto-Shivashinsky equation from highly noisy data. Reproducibility details can be found in the appendix.

**Korteweg-de Vries**  The Korteweg-de Vries equation describes waves in shallow water and is given by $u_t = u_{xxx} - uu_x$. Figure 2a shows the dataset: 2000 samples with 20% noise from a two-soliton solution. We compare our approach with I) Sparse Bayesian Learning with features calculated with numerical differentiation, II) a model discovery algorithm with PINNs, but non-differentiable variable selection called DeepMoD (Both & Kusters, 2021) and III) PDE-find (Rudy et al., 2017), a popular model discovery method for PDEs based on SINDy (Brunton et al., 2016). The first two benchmarks also act as an ablation study: method I uses the same regression algorithm but does not use a neural network to interpolate, while method II uses a neural network to interpolate but does not implement differentiable variable selection. In figure 2b and c we show that the differentiable approach recovers the correct equation after approximately 3000 epochs. Contrarily, DeepMoD recovers the wrong equation. Performing the inference 10 times with different seeds shows that the fully-differentiable approach manages to recover the Korteweg-de Vries equation nine times, while DeepMoD recovers the correct equation only twice - worse, it recovers the same wrong equation the other 8 times. Neither PDE-find nor SBL with numerical differentiation is able to discover the Korteweg-de Vries equation from this dataset, even at 0% noise due to the data sparsity.

---

[2]We choose to plot the losses of the original PINN loss $\mathcal{L}_{\text{data}}$ and $\mathcal{L}_{\text{reg}}$ because these are more easily interpreted than the likelihood-based losses we have introduced.

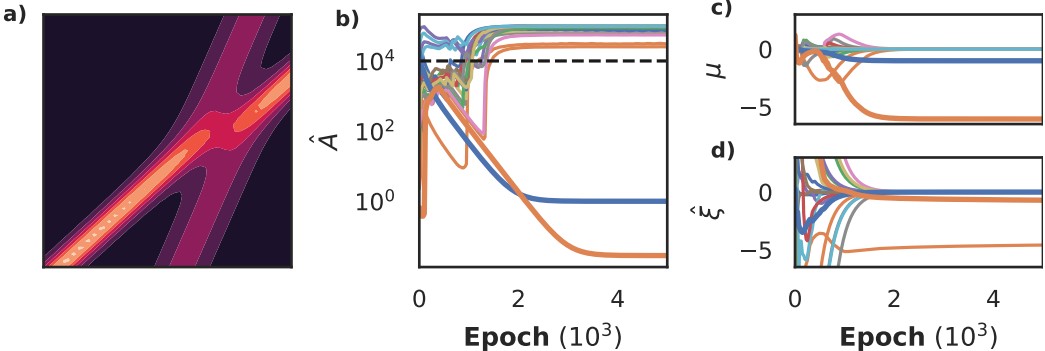

Figure 2: Comparison of a differentiable SBL-constrained model and an non-differentiable OLS-constrained model on a Korteweg-de Vries dataset (panel **a**) with a library consisting of 4th order derivatives and 3rd order polynomials, for a total of 20 candidate features. In panel **b** and **c** we respectively plot the inferred prior $\hat{A}$ and the posterior coefficients $\mu$. In panel **d** we show the non-differentiable DeePyMod approach. In panels **b** and **c** we see that the correct equation (bold blue line: $u_{xx}$, bold orange line: $uu_x$) is discovered early on, while the non-differentiable model (panel **d**) selects the wrong terms.

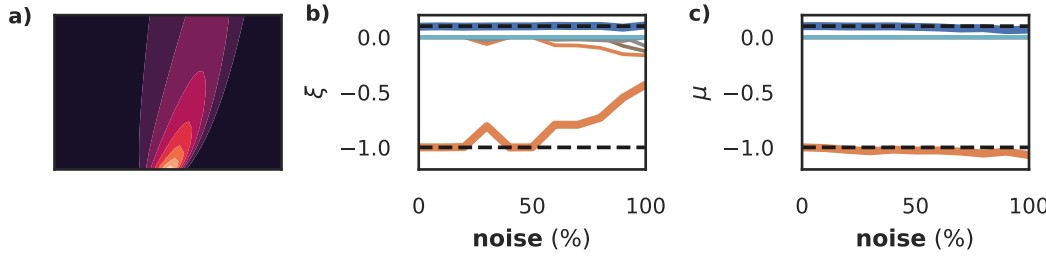

Figure 3: Exploration of robustness of SBL-constrained model for model discovery for the Burgers equation (panel **a**). We show the discovered equation over a range of noise for DeepMoD (panel **b**) and the approach presented in this paper (panel **c**). The bold orange and blue lines denotes $u_x x$ and $uu_x$, and black dashed line their true value.

**Burgers** We now explore how robust the SBL-constrained PINN is with respect to noise on a dataset of the Burgers equation, $u_t = \nu u_{xx} - uu_x$ (figure 3a)). We add noise varying from 1% to 100% and compare the equation discovered by benchmark method II (DeepMoD, panel b) and our approach (panel c) - the bold orange and blue lines denote $u_{xx}$ and $uu_x$ respectively, and the black dashed line their true value. Observe that DeepMoD discovers small additional terms for $> 50\%$ noise, which become significant when noise $> 80\%$. Contrarily, our fully differentiable approach discovers the same equation with nearly the same coefficients across the entire range of noise, with only very small additional terms ($\mathcal{O}(10^{-4})$). Neither PDE-find nor SBL with numerical differentiation is able to find the correct equation on this dataset at 10% noise or higher.

**Kuramoto-Shivashinsky** The Kuramoto-Shivashinksy equation describes flame propagation and is given by $u_t = -uu_x - u_{xx} - u_{xxxx}$. The fourth order derivative makes it challenging to learn with numerical differentiation-based methods, while its periodic and chaotic nature makes it challenging to learn with neural network based methods (Both et al., 2021). We show here that using the SBL-constrained approach we discover the KS-equation from only a small slice of the chaotic data (256 in space, 25 time steps), with 20% additive noise. We use a tanh-activated network with 5 layers of 60 neurons each, and the library consists of derivatives up to 5th order and polynomials up to fourth order for a total of thirty terms. Additionally, we precondition the network by training without the constraint for 10k epochs.

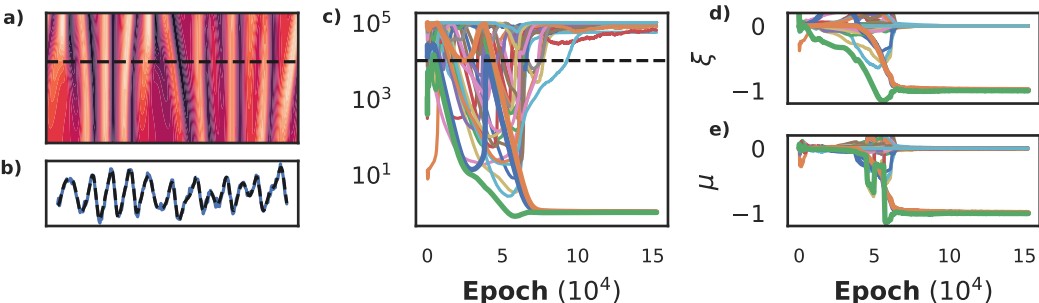

Figure 4: Recovering the Kuramoto-Shivashinsky equation. We show the chaotic data and a cross section in panels **a** and **b**. The periodicity makes this a challenging dataset to learn, requiring 200k iterations to fully converge before it can be recovered (panel **c**). Panels **d** and **e** show that the posterior and MLE of the coefficients yield nearly the same coefficients, indicating that the network was able construct an extremely accurate approximation of the data.

Training this dataset to convergence takes significantly longer than previous examples, as the network struggles with the data's periodicity (panel b). After roughly 70k epochs, a clear separation between active and inactive terms is visible in panel c, but it takes another 30k epochs before all inactive terms are completely pruned from the model. Panels d and e show the corresponding posterior and the maximum likelihood estimate of the coefficients using the whole library. Remarkably, the MLE estimate recovers the correct coefficients for the active terms, while the inactive terms are all nearly zero. In other words, the accuracy of the approximation is so high, that least squares identifies the correct equation.

### 4.3 Model discovery with normalizing flows

Consider a set of particles whose movement is described by a micro-scale stochastic process. In the limit of many of such particles, such processes can often be described with a deterministic macro-scale density model, determining the evolution of the density of the particles over time. For example, a biased random walk can be mapped to an advection-diffusion equation. The macroscale density models are typically more insightful than the corresponding microscopic model, but many (biological) experiments yield single-particle data, rather than densities. Discovering the underlying equation thus requires first reconstructing the density profile from the particles' locations. Classical approaches such as binning or kernel density estimation are either non-differentiable, non-continuous or computationally expensive. Normalizing Flows (NFs) have emerged in recent years as a flexible and powerful method of constructing probability distribution, which is similar to density estimation up to a multiplicative factor. In this section we use physics informed normalizing flows to learn a PDE describing the evolution of the density directly from unlabelled single particle data.

Since the conditional normalizing flow is used to construct the density, a precision denoting the noise level does not exist, and instead we set as prior for $\beta$ ($a = N, b = N \cdot 10^{-5}$). We consider a flow consisting of ten planar transforms (Rezende & Mohamed, 2015) and a hyper-network of two layers with thirty neurons each. The dataset consists of 200 walkers on a biased random walk for 50 steps, corresponding to an advection-diffusion model, with an initial condition consisting of two Gaussians, leading to the density profile shown in figure 5a. The two smallest terms in panel e correspond to the advection (bold green line) and diffusion (bold red line) term, but not all terms are pruned. Panels b, c and compare the inferred density (red line) to the true density (dashed black line) and the result obtained by binning. In all three panels the constrained NF is able to infer a fairly accurate density from only 200 walkers. We hypothesise that the extra terms are mainly due to the small deviations, and that properly tuning the prior parameters and using a more expressive transformation would prune the remaining terms completely. Nonetheless, this shows that NF flows can be integrated in this fully differentiable model discovery framework.

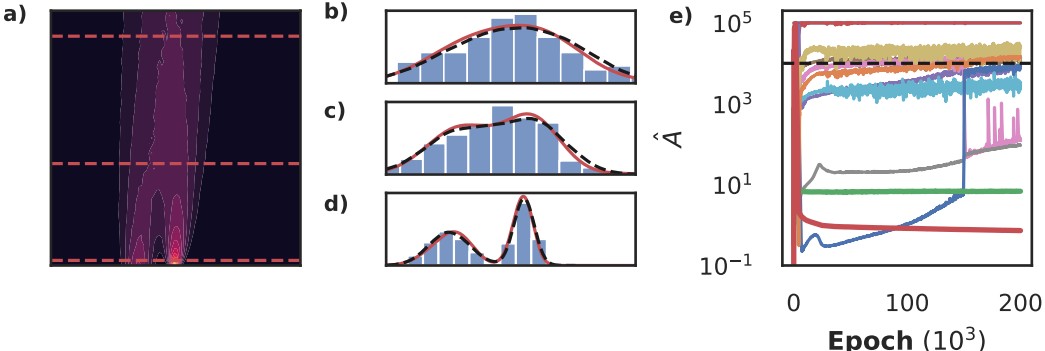

Figure 5: Using a tempo-spatial Normalizing Flow constrained by Sparse Bayesian Learning to discover the advection-diffusion equation directly from single particle data. Panel **a** shows the true density profile, and in panels **b c** and **d** we show the density inferred by binning (blue bars), inferred by NF (red) and the ground truth (black, dashed) at $t = 0.1, 2.5, 4.5$. Note that although the estimate of the density is very good, we see in panel **e** that we recover two additional terms (bold blue line: $u_x$, bold orange line $u_{xx}$.

## 5 OUTLOOK

Our experiments show a strong improvement over non-differentiable comparable methods, and opens up several new avenues to explore. One direction is the choice of the prior parameters for the precision. We presented a reasonable choice of prior parameters, but future work could find better estimates, for example a 'prior-scheduler', similar to a learning rate scheduler, or explore approaches to multitask learning. A different direction is exploring different Bayesian regression methods. For example, using a Laplacian (Helgøy & Li, 2020) or spike-and-slab prior can improve sparsity (Nayek et al., 2020). Alternatively, the prior can be used to introduce more structure into the problem. For example, the group-SBL could be used to combine data from several experiments (Babacan et al., 2014).

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

## A    REPRODUCIBILITY

**Optimizer**    We use the Adam optimizer with learning rate $2 \cdot 10^{-3}$ and $\beta = (0.99, 0.99)$ for all experiments.

**Figure 1**  We use a two soliton solution of the Korteweg-de Vries equation, starting at $x_0 = (-5.0, 0.0)$ with speeds $c = (5, 2)$ . We regularly sample 50 points on the spatial domain $[-6, 7]$ and 40 along the temporal domain $[0.1, 3.0]$, and add $10\%$ (in terms of the std. of the data) white noise. The network is composed of 3 hidden layers of 30 neurons. As library we derivatives up to fourth order, polynomials up to third and all combinations of these.

**Figure 2**  We use a two soliton solution of the Korteweg-de Vries equation, starting at $x_0 = (-5.0, 0.0)$ with speeds $c = (5, 2)$ . We regularly sample 50 points on the spatial domain $[-6, 7]$ and 40 along the temporal domain $[0.1, 3.0]$, and add $20\%$ (in terms of the std. of the data) white noise. The network is composed of 3 hidden layers of 30 neurons. As library we derivatives up to fourth order, polynomials up to third and all combinations of these.

**Figure 3**  We solved the Burgers equation for a delta peak initial condition, regularly sample 50 points on the spatial domain $[-3, 4]$ and 20 along the temporal domain $[0.5, 5.0]$, and add $1 : 10 : 100\%$ (in terms of the std. of the data) white noise. The network is composed of 3 hidden layers of 30 neurons. As library we derivatives up to fourth order, polynomials up to third and all combinations of these.

**Figure 4**  We numerically solve the Kuramoto-Shivashinsky equation for a random initial condition with a spatial resolution of 1024 and spatial resolution of 256 frames. We take a slice of data between $45 < t55$ (25 frames) and subsample every 4th point along the spatial axis (giving 256 points / frame) and add $20\%$ (in terms of the std. of the data) white noise. The network is composed of 5 hidden layers of 60 neurons. As library we derivatives up to fifth order, polynomials up to fourth and all combinations of these.

**Figure 5**  We release 200 random particles on a biased random walk with $D = 1.5, v = 0.5$ from an initial distribution of two Gaussians centred at $x_0 = (-5, -1)$ with widths $\sigma = (1.5, 0.5)$ and take 50 steps with $\Delta t = 0.05$. The hyper network is composed of 2 hidden layers of 30 neurons and we use a NF of 10 layers. As library we derivatives up to third order, polynomials up to second and all combinations of these.

