# OpenReview forum: "Fully differentiable model discovery"
_ICLR.cc/2022/Conference — ICLR 2022 Submitted_

### Official Review · Reviewer_MdGD · 2021-10-31

**Correctness:** 2
**Technical Novelty And Significance:** 2
**Empirical Novelty And Significance:** 3
**Recommendation:** 5
**Confidence:** 3

**Main Review:**

This work proposes to denoise noisy data with PINNs, and use SBL on (fixed basis expansions of) the denoised data to provide differentiable basis selection.  The broad idea of combining SBL-like procedures with deep models is natural and has been examined in the context of generative modeling (e.g. Dai and Wipf, 2019), but application to PDE learning appears new.  Numerical experiments seem to demonstrate some improvements, although I don't work on PDE learning so am not completely sure.

My main concern is about the optimization landscape.  For SBL to extract truly sparse coefficients we need its objective to go to infinity; but the PINN loss is a normal likelihood, and will be bounded by the noise level of the data, which is assumed to be finite in this setting (this is different from typical sparse learning scenarios where a low level of noise is assumed).  Therefore, optimizing the sum of the two objectives seems a bad idea.  In particular, one "global optima" would be for the PINN to estimate $\hat u \equiv 0$, incurring a finite reconstruction error, so that the SBL loss tends to negative infinity.  The authors seem to work around this issue by imposing proper hyperpriors, and pretrain the PINN so that the joint optimization starts at an informative initial value, these fixes seem somewhat hackish: proper hyperpriors will restrict the algorithm's ability to obtain truly sparse solutions in practice, or the behavior of the algorithm may depend on the value of the hyperprior parameters $a,b,c,d$ which are difficult to determine a priori.

Also, while I don't work on PINNs so cannot judge the originality, a quick search leads to this work which should be discussed:

Amuthan A. Ramabathiran and Prabhu Ramachandran^, "SPINN: Sparse, Physics-based, and partially Interpretable Neural Networks for PDEs", Journal of Computational Physics, Volume 445, pages 110600, 2021.

Minor questions/comments:

* In the first experiment, why does SBL without PINN fail in the noiseless setting? Is this due to numerical errors in computing higher-order derivatives?
* $\mathbf{u}_t$(=$\partial_t \hat{\mathbf{u}}$?) isn't defined in the text. The choices of notation might also be optimized.
* Formatting issues: better to use `\eqref` for equations, `Eq.~` as opposed to `Eq. `, etc.

References:

Dai and Wipf (2019), Diagnosing and Enhancing VAE Models, in ICLR.

**Summary Of The Paper:**

This work proposes to combine sparse-Bayesian learning (SBL) with physics informed neural networks (PINNs) to achieve feature/basis selection in learning PDEs.

**Summary Of The Review:**

Pros:
+ The problem of differentiable model discovery on noisy data seems important, and the proposed method appears promising and easy to implement

Cons:
- Optimization landscape appears insensible

**Post-rebuttal update**: while the authors clarified on the experiments and related work, my core concern on the optimization landscape remains.  Therefore I am keeping my score unchanged.

---

> ### Author Response · Authors · 2021-11-17
> **Response to reviewer.**
>
> Thank you for comments. We respond to them point-wise below:
>
> **My main concern is about the optimization landscape. For SBL to extract truly sparse coefficients we need its objective to go to infinity; but the PINN loss is a normal likelihood, and will be bounded by the noise level of the data, which is assumed to be finite in this setting (this is different from typical sparse learning scenarios where a low level of noise is assumed). Therefore, optimizing the sum of the two objectives seems a bad idea. In particular, one "global optima" would be for the PINN to estimate, incurring a finite reconstruction error, so that the SBL loss tends to negative infinity. The authors seem to work around this issue by imposing proper hyperpriors, and pretrain the PINN so that the joint optimization starts at an informative initial value, these fixes seem somewhat hackish: proper hyperpriors will restrict the algorithm's ability to obtain truly sparse solutions in practice, or the behavior of the algorithm may depend on the value of the hyperprior parameters which are difficult to determine a priori.**
>
> This is a valid concern and there's a lot of active research centering around this topic. We first want to clarify a few points:
>
> - While the prior inverse width $\alpha_i$ goes to infinity, the marginal likelihood, i.e. the loss of the SBL does not. Numerically dealing with infinities is impossible, and we define a numerical cutoff after which alpha is considered infinite (10^5), similar to the scikit-learn implementation.
> - The constraint term in a PINN becomes small, but not exactly zero (i.e. we approximately satisfy the physics). Analogously, the noise term of the SBL becomes small but not infinite, and the likelihood does not go to infinity.
> - We do not just sum these two terms - our loss function is derived from a multi-task probabilistic perspective.
> - We only pretrain the network for the Kuramoto-Shivashinsky equation, as PINNs have trouble converging in this dataset.
>
> Indeed, without the correct hyperprior the SBL gets optimized first and traps the model in a local minimum. However, we derive a reasonable estimate and while further research is required into better choices, our estimate yields excellent results on three different datasets across various noise levels. We are thus unsure what the reviewer means by 'proper hyperpriors will restrict the algorithm's ability to obtain truly sparse solutions in practice', as our experiments obtain positive result, and ask for clarification. To summarize, the concern about the loss landscape is valid, but we show that in practice this is not a big impediment.
>
> **Also, while I don't work on PINNs so cannot judge the originality, a quick search leads to this work which should be discussed:**
>
> > Amuthan A. Ramabathiran and Prabhu Ramachandran^, "SPINN: Sparse, Physics-based, and partially Interpretable Neural Networks for PDEs", Journal of Computational Physics, Volume 445, pages 110600, 2021.
>
>  The SPINN paper considers solving PDEs by coupling a neural network into a radial basis function layer. This work 1) does not discover PDEs, 2) is only sparse in the sense that 'we use less parameters than a dense NN' and 3) is only interpretable in the shape of the kernel functions. We ask the reviewer to expand on why they feel it should be discussed, as we do not see its relevance to our work.
>
> **In the first experiment, why does SBL without PINN fail in the noiseless setting? Is this due to numerical errors in computing higher-order derivatives?**
>
> Exactly! The error of numerical differentiation intrinsically scales with the sampling distance, and once the distance is too large (as in the dataset we present), even on a noiseless dataset model discovery with numerical differenitation becomes impossible. This is our main argument for the use of neural network interpolators for model discovery.
>
> **$u_t = \partial_t u$ isn't defined in the text. The choices of notation might also be optimized - Formatting issues: better to use \eqref for equations, Eq.~ as opposed to Eq. , etc.**
>
> Thank you for bringing this to our attention, we will fix this for the camera ready version.

---

> > ### Comment · Reviewer_MdGD · 2021-11-20
> > **Response**
> >
> > Thanks for your response, especially around the reference and experiments; but my concern about the landscape remains:
> >
> > You state that the marginal likelihood does not go to infinity because you defined a numerical cutoff.  But my concern is not about the numerical implementation; it is about the fact that combining the marginal likelihood objective, which can be extremely large, with a bounded reconstruction error term is dangerous and in principle may lead to nonsensical global optima.  The marginal likelihood can (and needs to) be very large because (to my knowledge) theoretical analyses of SBL relies on it going to infinity (e.g. Wipf and Rao, 2004).  Your reconstruction error (12) should be bounded because you assume a non-zero level of noise.
> >
> > Combining the two points above, we arrive at the $\hat u\equiv 0$ example in my original review.  To avoid it, *in principle*, you will have to use a very aggressive truncation level, which would prevent provable recovery of sparse components, or else the (truncated) marginal likelihood will be a large albeit finite value at my example, creating a useless global optima.  Of course the final algorithm works in practice which is a nice thing, but this theoretical issue is concerning, especially because it is unique to your setting of recovering sparse components under non-negligible noise.  This makes the use of SBL here a lot less appealing than potential alternatives (e.g. spike-and-slab prior, L1 regularization, etc).
> >
> > > We are thus unsure what the reviewer means by 'proper hyperpriors will restrict the algorithm's ability to obtain truly sparse solutions in practice', as our experiments obtain positive result, and ask for clarification.
> >
> > "In practice" should have been replaced by "in theory"; sorry for the typo.  And this is because, to my knowledge, theoretical analyses of SBL assumes improper prior, as they rely on marginal likelihood going to infinity (e.g. Wipf and Rao, 2004).  The original work of Tipping (2001) also focused on the improper prior.
> >
> > References:
> >
> > Wipf and Rao (2004), Sparse Bayesian Learning for Basis Selection, IEEE Transactions on Signal Processing.

---

### Official Review · Reviewer_F5HG · 2021-11-02

**Correctness:** 3
**Technical Novelty And Significance:** 3
**Empirical Novelty And Significance:** 2
**Recommendation:** 5
**Confidence:** 3

**Main Review:**

**Strengths**

The approach presented in this work seems sample efficient (requires more experiments for empirical validation) and can identify equations from noisy datasets as compared to the baselines.

**Weaknesses**

* The authors need to compare their approach against PDE-NET 2.0 [1]. I'd be surprised if PDE-NET 2.0 doesn't perform competitively with the proposed method.

* The authors need to present the exact equations retrieved by all the 3 baselines for each experiment. Also, they need to show which coefficients the colored lines correspond to in the legend (Fig. 2(c), 2(d), 2(e), 3(e)) to improve the readability of the plots.

* I'm not sure if I understand the concerns raised by the authors regarding the regularisation constant in $l_1$ regularization in the following statement in the Introduction - *Applying $l_1$ regularisation can alleviate this problem, but raises the question of how strongly
to apply it*. $\lambda$, the strength of the regularization constant, is a hyperparameter that is varied uniformly. And the model/equation with the least MSE (or any other relevant metric) is picked.

* The authors need to elaborate on the training strategy they used for DeepMod. DeepMod follows a similar learning strategy as [1] (i.e. use L1 regularization to induce sparsity into the Neural Network) to identify concise equations. In [2] the authors have demonstrated that such a learning scheme is susceptible to random restarts. This is also consistent with the observation made by the authors in sec.4.2. For that reason, a grid search has to be performed for different values of $\lambda$ (strength of $l_1$ regularization) and $seeds$. Then the correct model/equation needs to be selected from these candidates using some metric, for instance in [2] the authors used the extrapolation error. Thus model selection is really important and if there doesn't exist a robust model selection strategy that can find the right equation from the pool of candidate equations then finding the right equation just once isn't at all useful. Thus, I'd like for the authors to shed some light on the training curriculum they followed for DeepMod. I'm assuming they didn't run DeepMod with a single set of hyperparameters and thus elaborate on the model selection strategy they used.

* The authors have shown that their approach is able to identify the correct equations for a very limited number of cases like - Korteweg-de Vries, Burgers, Kuramoto-Shivashinsky, and Normalizing flows. They need to demonstrate that their algorithm can identify a variety of equations. Thus I strongly suggest that they generate PDEs with random terms and coefficients and report the number of cases for which the network succeeds or fails to identify the correct equation as they did in [2].

* (Optional) For the above experiment it'd really insightful to see how training data size affects the quality of the learned equation.

**References**

[1] Z. Long, Y. Lu, X. Ma, and B. Dong. PDE-Net: Learning PDEs from Data. ArXiv e-prints, 2017

[2] Sahoo S, Lampert C, Martius G. 2018 Learning equations for extrapolation and control. In Proc. of the 35th Int. Conf. on Machine Learning, Stockholm, Sweden, 10–15 July 2018 (eds J Dy, A Krause), vol. 80, pp. 4442–4450. Proceedings of Machine Learning Research


**Summary Of The Paper:**

becauseThis work presents a model discovery algorithm to learn the PDEs from data. Their method uses Sparse Bayesian Learning to identify a minimal set of terms (along with their coefficients) from a set of predefined terms that constitute the PDE. Finally, they demonstrate that their algorithm can identify the PDE equations precisely for the following problems -  Korteweg-de Vries, Burgers, Kuramoto-Shivashinsky, and Normalizing flows.

**Summary Of The Review:**

The only reason I feel the paper is marginally below the acceptance threshold is that -
* The authors haven't compared their method against one of the most popular methods - PDE-NET 2.0
* Lack of extensive experiments (see above) to demonstrate that their approach can identify a variety of equations.

---

> ### Author Response · Authors · 2021-11-18
> **Response to reviewer I.**
>
> Thank you for your comments. We respond to your concerns pointwise below:
>
> **The authors need to compare their approach against PDE-NET 2.0 [1]. I'd be surprised if PDE-NET 2.0 doesn't perform competitively with the proposed method.**
>
> Many different approaches exist, and unfortunately we cannot include all. We have chosen the mentioned baselines because they are either extremely popular (SINDY), or have shown a robustness to noise and dataset size (DeepMoD). Additionally, we have experience with both and each baseline acts as an ablation study for our work. We did not include PDE-NET as we have no experience with it and wish to prevent unfair comparisons. That said, we doubt if PDE-NET would perform competitively, as it relies on numerical differentiation, and we show this is a key component in constructing a performant algorithm.
>
> **The authors need to present the exact equations retrieved by all the 3 baselines for each experiment.**
>
>  We experimented with various ways to depict the found equations, but due to the size of the library were unable to do this clearly. Conversely, the baselines discovered equations consisting of many terms, which were not particularly insightful.
>
> **Also, they need to show which coefficients the colored lines correspond to in the legend (Fig. 2(c), 2(d), 2(e), 3(e)) to improve the readability of the plots.**
>
> We agree and will change this for the camera-ready version.
>
> **I'm not sure if I understand the concerns raised by the authors regarding the regularisation constant in
> regularization in the following statement in the Introduction - Applying $l_1$ regularisation can alleviate this problem, but raises the question of how strongly to apply it., the strength of the regularization constant, is a hyperparameter that is varied uniformly. And the model/equation with the least MSE (or any other relevant metric) is picked.**
>
> Reviewer rgZy asks a similar question- why not simply apply a sparsity promoting regularisation to the coefficients of $\Theta$ and get rid of the mask? Doing so would require to pick a value of $\lambda$ and train the network with this value, and to pick the right value of the hyperparameter would thus require retraining the network for each value of $\lambda$, a computationally expensive approach. In this work we effectively learn the strength of the regularisation during training, thus not needing to train several networks.
>
> The rest of your questions are answered in the next response.

---

> > ### Author Response · Authors · 2021-11-18
> > **Response to reviewer, II.**
> >
> >
> > **The authors need to elaborate on the training strategy they used for DeepMod. DeepMod follows a similar learning strategy as [1] (i.e. use L1 regularization to induce sparsity into the Neural Network) to identify concise equations. In [2] the authors have demonstrated that such a learning scheme is susceptible to random restarts. This is also consistent with the observation made by the authors in sec.4.2. For that reason, a grid search has to be performed for different values of (strength of regularization) and then the correct model/equation needs to be selected from these candidates using some metric, for instance in [2] the authors used the extrapolation error. Thus model selection is really important and if there doesn't exist a robust model selection strategy that can find the right equation from the pool of candidate equations then finding the right equation just once isn't at all useful. Thus, I'd like for the authors to shed some light on the training curriculum they followed for DeepMod. I'm assuming they didn't run DeepMod with a single set of hyperparameters and thus elaborate on the model selection strategy they used.**
> >
> > Indeed when using a neural network to model the equation (i..e the approach used by [2]) one is very dependent on the initialisation and multiple runs are required. However, DeepMoD does not use gradient descent and random initialisation for the constraint, and is thus are much less susceptible to this issue; the only randomness in our model is the initialisation from the neural network. The parameters of the constraint are then calculated deterministically from the approximation, and the strength of the regularisation can be determined using cross-validation without retraining the entire network. All this is detailed in the cited paper.
> >
> > **The authors have shown that their approach is able to identify the correct equations for a very limited number of cases like - Korteweg-de Vries, Burgers, Kuramoto-Shivashinsky, and Normalizing flows. They need to demonstrate that their algorithm can identify a variety of equations. Thus I strongly suggest that they generate PDEs with random terms and coefficients and report the number of cases for which the network succeeds or fails to identify the correct equation as they did in [2].**
> >
> > The experiments we present are the typical benchmarks used in model discovery algorithms.  We also take this opportunity to stress the difference with [2]: the authors of [2] perform symbolic regression. As such, their datasets are densely sampled and noiseless, their equations do not consist of derivatives and their goal is on recovering on long, complex equations. Contrarily, our goal is to discover PDEs from experimental (i.e. noisy and sparse) data; here the difficulty is not in the equation to be discovered, but in calculating the features with higher-order derivatives. As we show, baselines fail even at 0% noise due to the error induced by the numerical differentiation.  Comparing with [2] is, in our opinion, not a good comparison, as they essentially solve a different problem and have a different goal.
> >
> > **(Optional) For the above experiment it'd really insightful to see how training data size affects the quality of the learned equation.**
> >
> > We have done such an experiment for the Burgers equation, and observed that performance remained similar across a wide range of data size. We will include it in the camera ready version.

---

### Official Review · Reviewer_rgZy · 2021-11-03

**Correctness:** 3
**Technical Novelty And Significance:** 2
**Empirical Novelty And Significance:** 3
**Recommendation:** 5
**Confidence:** 3

**Main Review:**

The paper provides a nice introduction and motivation to the problem of identifying simple governing dynamical equations from observed data.

It seems like there are two main contributions to the method:
Incorporating a soft relaxation of a hard constraint (the discrete mask) used in previous work.
Using sparse Bayesian learning to formulate the sparse regression problem in a fully Bayesian way.
The paper seems to focus on (a), and does not focus on the benefits of (b) in the text.

First, I have a really basic question: in eq (3), why not add a sparsity penalty on the coefficients of $\Theta$ and then get rid of the mask? Would this not also be a soft relaxation of the hard mask constraint? This also seems more in line with prior work such as SINDy. I’m sure there’s some reason why this isn’t feasible, but I couldn’t figure out why.

The text describing the first experimental results (4.2) suggest that two baseline methods were tried, but Figure 2 only compares against one baseline method? What am I missing here?

In addition, more details could be provided about certain choices in the experiments. How sensitive are the results to different hyperparameter choices (choice of the network architecture, number of library terms, regularization hyperparameters, etc).

Some suggestions regarding the figures:
- Add axes labels to all figures. For example, in Figure 2a or 3a, what are the axes? In Figure 2b and 2c or 3b and 3c, it would be helpful if each panel had a short descriptive title.
- It would be helpful to see the recovered equations and reconstruction for at least one of the example applications
- Consider plotting the relative error of each parameter fit, instead of (or in addition to) the actual values (e.g. Fig 3b and 3c).

**Summary Of The Paper:**

This paper studies the problem of identifying sets of equations that govern the dynamics of observed natural systems. Building off of work that uses sparse regression to identify a small set of active terms from a large dictionary of possible terms, this paper proposes a soft relaxation of a hard constraint used for selecting out possible terms in the optimization problem. The paper applies this technique to recover the underlying equations from data generated by a few canonical PDEs.

**Summary Of The Review:**

Overall, I had a bit of a hard time understanding just how significant the improvements were when using this method. Is the differentiable method more robust to hyperparameters? In principle, with careful tuning, it seems like the non-differentiable version should also be to recover the same equations, given that they both use the same model for approximating the dynamics. In addition to significance, the clarity of the paper could also be improved.

---

> ### Author Response · Authors · 2021-11-17
> **Response to reviewer.**
>
> Thank you for the review. We respond to your comments below:
>
> **It seems like there are two main contributions to the method: Incorporating a soft relaxation of a hard constraint (the discrete mask) used in previous work. Using sparse Bayesian learning to formulate the sparse regression problem in a fully Bayesian way. The paper seems to focus on (a), and does not focus on the benefits of (b) in the text.**
>
> While technically speaking you are right in making the distinction between the two main contributions, essentially they are the same. The soft relaxation cannot be performed on the component-wise basis without a Bayesian approach: applying an approach such as cross validation to determine the regularization of each separate component is computationally too expensive. Only by using a Bayesian approach (in our case, SBL) can we calculate the component-wise regularization and apply the soft relaxation.
>
> **First, I have a really basic question: in eq (3), why not add a sparsity penalty on the coefficients of $\Theta$ and then get rid of the mask? Would this not also be a soft relaxation of the hard mask constraint? This also seems more in line with prior work such as SINDy. I’m sure there’s some reason why this isn’t feasible, but I couldn’t figure out why.**
>
> This isn't a basic question at all - in fact, it's the approach taken in the first [deepmod paper](https://arxiv.org/abs/1904.09406). While this works fairly well, there are several issues:
> 1) the l1 penalty is inconsistent, thus not always capable of yielding the ground truth,
> 2) the l1 penalty often yields several small but not exactly zero terms, requiring thresholding.
> 3) How to determine the strength of the regularization? Since its incorporated in the network, this would involve retraining the entire network, which is computationally expensive.
>
> **The text describing the first experimental results (4.2) suggest that two baseline methods were tried, but Figure 2 only compares against one baseline method? What am I missing here?**
>
> We use three baselines, as stated in the korteweg-de vries section:
>
> > We compare our approach with I) Sparse Bayesian Learning with features calculated with numerical differentiation, II) a model discovery algorithm with PINNs, but non-differentiable variable selection called DeepMoD (Both & Kusters, 2021) and III) PDE-find (Rudy
> et al., 2017), a popular model discovery method for PDEs based on SINDy (Brunton et al., 2016)."
>
> but indeed we only show approach II) in figure 2. Approaches I and III do not require training a neural network and can thus not be shown as figure 2b-d, and we have made the choice to discuss these baselines only in the text, at the end of the section:
>
> > Neither PDE-find nor SBL with numerical differentiation is able to discover the Korteweg-de Vries equation from this dataset, even at 0% noise due to the data sparsity.
>
> We shall clarify this in the text.
>
> **In addition, more details could be provided about certain choices in the experiments. How sensitive are the results to different hyperparameter choices (choice of the network architecture, number of library terms, regularization hyperparameters, etc).**
>
> PINNs are not known to yield different results across network sizes, but we agree that an experiment showing the dependence will strengthen our work - we shall add it to the camera-ready version. We use libraries of sizes much larger than typical sparse regression approaches, so we essentially cover the whole space of number of library terms already; models with 6th order derivatives do no exist to our knowledge, nor do models with polynomials of order >5, so making the library larger is not of physical interest. The only other hyperparameter is the choice of the hyperprior, for which we derive a reasonable choice which is dependent on the data and which is used with succes in all three experiments.
>
> **Some suggestions regarding the figures:
> Add axes labels to all figures. For example, in Figure 2a or 3a, what are the axes? In Figure 2b and 2c or 3b and 3c, it would be helpful if each panel had a short descriptive title.
> It would be helpful to see the recovered equations and reconstruction for at least one of the example applications
> Consider plotting the relative error of each parameter fit, instead of (or in addition to) the actual values (e.g. Fig 3b and 3c).**
>
> Thank you for the suggestions - we will make these changes in the camera-ready version.

---

### Official Review · Reviewer_VxhG · 2021-11-06

**Correctness:** 3
**Technical Novelty And Significance:** 2
**Empirical Novelty And Significance:** 2
**Recommendation:** 5
**Confidence:** 3

**Main Review:**

Pros:
  - The task of learning a PDE is interesting.

Cons:
  - But the technical aspect ultimately becomes sparse regression.
  - There are many other differentiable alternatives that are simpler, or newer, that are not explored.

Comments:
  - This is an arguably straightforward combination of prior works. As such, I'd liked to have seen more analysis on why the chosen approach (SBL) is good. For instance, there are many better sparsity-inducing priors than a Gaussian, such as the horseshoe prior, log-uniform, and other "spike-and-slab" style distributions. See for instance, [1, 2]. A comparison of such approaches would provide more insight.

  - The baselines warrant more description, as I'm not sure why they fail. For instance, the main alternative method seems to be a greedy search over the mask (after Eq 3: "The mask is updated periodically by some sparse regression technique, and as terms are pruned, the constraint becomes stricter,..."). But more sophisticated search methods should fare better.

  - Stronger baselines. For instance, before relaxing the mask, what about using REINFORCE gradient to learn binary masks? What about using the Gumbel-softmax or other types of relaxations first? (I imagine a sparsity-inducing approach should still fare better overall, but a comparison to alternative/simpler approaches should be warranted.) There are many other differentiable approaches to sparse regression that the paper does not mention. For a paper titled "Fully differentiable model discovery", I'd expect more exploration among these methods and ultimately a convincing conclusion on why the chosen approach seems the most worthwhile. (Is SBL with the chosen prior particularly suited for PINNs in any way?)

  - PINF: I'm quite lost. (i) Is there's a vector field u(t, x) that is in the PINF, for which the SBL loss is applied on? (ii) I didn't understand the motivation behind a PINF, is it because the Gaussian observation model in PINN may be insufficient? (iii) Perhaps an example would help; what is the underlying PDE for the random walk experiment?

  - Why optimize A and beta all the way, and not jointly with the PINN parameters?

Clarity issues:

  - The paper mentions that this approach can provide "a rigorous criterion for deciding whether a term is active or not". What is the criterion? And why is it "rigorous". Such approach should still requires a threshold for deciding whether a term is active if you ultimately want a deterministic value for the mask.

  - At the top of page 5, "our method does not require backpropagating through the solver" -> What solver is this referring to? Is this referring to optimizing A and beta? (By the way, since there's no closed form expression for A and beta, why not optimize for them at the same time as theta, which should be faster overall?)

  - How does the DeepMoD method differ from what is written in/after Eq 3?

  - When noise is added, is this being added to u only? What if you add noise to x or t?

  - What exactly is 1% or 100% noise? What is this a percentage of? (i.e. does it correspond to some maximum standard deviation for some gaussian noise distribution?)

Minor:

  - Under Eq 18: what is little "n"? Should that be N?

  - Figure 2: what is OLS? ordinary least squares? Is this referring to DeepMoD?

  - Figure 2b: what is the dashed line?

  - Is it DeePyMod or DeepMoD? Both names seem to be used in the paper.

  - Maybe consider providing in-line labels (i.e. display u_x, u_xx, etc on top of the line), especially for Figure 5e.

[1] "Handling Sparsity via the Horseshoe" Carvalho et al. (2009)

[2] "Bayesian Compression for Deep Learning" Louizos et al. (2017)

**Summary Of The Paper:**

This paper proposes learning an underlying PDE through the use of a physics-informed neural network and sparsity regularization on the partial derivatives. A neural network representing the PDE solution is trained simultaneously with a sparse regression on the partial derivatives of this neural network.

The main novelty seems to be combining the model discovery loss of Both et al. 2021 with a sparse prior, relaxing the objective and allowing gradient-based optimization.

The paper also discusses physics-informed normalizing flows but I feel like it's a bit disconnected / orthogonal to the rest of the paper and not enough information is given; I did not understand the motivation, how it connects to a PDE solution, and how it's applied in the experiments.

**Summary Of The Review:**

This paper explores only one of many differentiable approaches to learning binary variables. I think a more comprehensive comparison against alternative differentiable approaches, from simple REINFORCE to horseshoe priors, would provide more insight and ultimately a convincing conclusion on why the chosen approach seems the most worthwhile. I do not understand the relevance of the PINF section; perhaps it could be its own standalone paper instead. There are also some clarity issues.

---

> ### Author Response · Authors · 2021-11-18
> **Respons to reviewer, I.**
>
> Thank you for your comments, we respond to them below in two comments.
>
> **But the technical aspect ultimately becomes sparse regression.**
>
> Our approach hinges on the combination of sparse regression with a neural network based interpolator, and while sparse regression indeed might be 'simple', it is widely used because it yields great results .
>
> **There are many other differentiable alternatives that are simpler, or newer, that are not explored.**
>
> A key difference with other differentiable masking approaches is that we are looking for a very specific mask - that corresponding to the equation. If only one extra term is present, we are discovering the wrong equation. This makes model discovery using much more challenging than the usecases in which masking is typically applied. We ask the reviewer to clarify which alternatives they have in mind, as to our knowledge these hinge on defining a certain sparsity level (which does not exist for us as we don't know the equation).
>
> **This is an arguably straightforward combination of prior works.  As such, I'd liked to have seen more analysis on why the chosen approach (SBL) is good. For instance, there are many better sparsity-inducing  priors than a Gaussian, such as the horseshoe prior, log-uniform, and  other "spike-and-slab" style distributions. See for instance, [1, 2]. A  comparison of such approaches would provide more insight.**
>
> The message of our work is to show how a neural network based interpolator with a constraint (i.e. a PINN) can be combined with learning the constraint using a Bayesian method. Indeed, we mention in our work that different, more sparsity-inducing priors can be used to increase performance. We used the SBL here as it is well-known and can be analytically solved (i.e. no need to sample), making our work much more easily interpretable.
>
> **The baselines warrant more description, as I'm not sure why they  fail. For instance, the main alternative method seems to be a greedy  search over the mask (after Eq 3: "The mask is updated periodically by  some sparse regression technique, and as terms are pruned, the  constraint becomes stricter,..."). But more sophisticated search methods should fare better.**
>
> **Stronger baselines. For instance, before relaxing the mask, what  about using REINFORCE gradient to learn binary masks? What about using  the Gumbel-softmax or other types of relaxations first? (I imagine a  sparsity-inducing approach should still fare better overall, but a  comparison to alternative/simpler approaches should be warranted.) There are many other differentiable approaches to sparse regression that the  paper does not mention. For a paper titled "Fully differentiable model  discovery", I'd expect more exploration among these methods and  ultimately a convincing conclusion on why the chosen approach seems the  most worthwhile. (Is SBL with the chosen prior particularly suited for  PINNs in any way?)**
>
> We used these baselines because they are used in the model discovery community. Approaches such as the REINFORCE gradient or Gumbel-softmax are new to us and seem worthy of investigation, but have never been used to perform model discovery and hence we didn't include them.
>
> **PINF: I'm quite lost. (i) Is there's a vector field u(t, x) that  is in the PINF, for which the SBL loss is applied on? (ii) I didn't  understand the motivation behind a PINF, is it because the Gaussian  observation model in PINN may be insufficient? (iii) Perhaps an example  would help; what is the underlying PDE for the random walk experiment?**
>
> In the case of the biased random walk experiment we present, the underlying PDE is an advection-diffusion model. However, this equation models the density of walkers and we only observe the position of single walkers. To recover this PDE requires first to construct the density from the walkers, and then perform model discovery on this density. This density is rarely Gaussian, and we use a Normalizing flow as a  differentiable time-dependent density estimator in lieu of classical approaches such as KDE or binning.

---

> ### Author Response · Authors · 2021-11-18
> **Reponse to reviewer, II.**
>
>
>
> **At the top of page 5, "our method does not require  backpropagating through the solver" -> What solver is this referring  to? Is this referring to optimizing A and beta? (By the way, since  there's no closed form expression for A and beta, why not optimize for  them at the same time as theta, which should be faster overall?)**
> **Why optimize A and beta all the way, and not jointly with the PINN parameters?**
>
> We mention in the section 'removing free variables' that optimizing these parameters every step yields faster convergence as the constraint , and makes the influence of random initialisation much smaller (see response to reviewer F5HG). The solver is indeed referencing the optimization used for A and beta.
>
> **The paper mentions that this approach can provide "a rigorous  criterion for deciding whether a term is active or not". What is the  criterion? And why is it "rigorous". Such approach should still requires a threshold for deciding whether a term is active if you ultimately  want a deterministic value for the mask.**
>
> If the prior $\alpha_i$ goes to infinity, the corresponding term can be pruned. Numerically speaking we follow scikitlearn and consider a term zero if $\alpha > 10^5$, i.e. this is a numerical criterion.
>
> **What exactly is 1% or 100% noise? What is this a percentage of?  (i.e. does it correspond to some maximum standard deviation for some  gaussian noise distribution?)**
>
> **How does the DeepMoD method differ from what is written in/after Eq 3? When noise is added, is this being added to u only? What if you add noise to x or t?**
>
> We add noise only to $u$, with the percentage being of the standard deviation of the dataset. Adding noise to $x$ and $t$ has not been studied to our knowledge and falls out of the scope of this paper.
>
> **Minor: Under Eq 18: what is little "n"? Should that be N?**
>
> Indeed, thank you for bringing this to our attention.
>
> **Figure 2: what is OLS? ordinary least squares? Is this referring to DeepMoD?**
>
> Indeed this refers to OLS and DeepMoD. We shall clarify this in the camera-ready version.
>
> **Figure 2b: what is the dashed line?**
>
> This refers to the noise level, i.e. the value the MSE should go to. We shall clarify this in the next version.
>
> **Is it DeePyMod or DeepMoD? Both names seem to be used in the paper.**
>
> DeePyMoD is the software package implementing DeepMoD. We shall fix the inconsistency.

---

### Decision · Program_Chairs · 2022-01-20

**Decision:**

Reject

**Comment:**

The reviewers are in consensus that this manuscript falls just short of the bar. I recommend that the authors take their recommendations into consideration in revising their manuscript, with a particular focus on comparison to the state of the art.